

**Multiply factors driving continual post-wildfire debris flows with**
**varied rainfall thresholds in the Reneyong Valley, southwestern**
**China**
Mingfeng Deng[1,2], Yong Zhang[1,2], Mei Liu[1,2], Yuanhuan Wang[3], Wanyin Xie[3], and Ningsheng
Chen[1]*
([1] Key Laboratory of Mountain Hazards and Surface Process, Institute of Mountain Hazards and Environment,
Chinese Academy of Sciences, Chengdu 610041, P R. China;
[2] University of Chinese Academic of Sciences, Beijing 100049, China
[3] Sichuan Institute of Geological Engineering Investigation, Chengdu, 610072, P R. China)
* Corresponding author: chennsh@imde.ac.cn
**Abstract**: In early June of 2014, wildfire struck the Reneyong Valley in the central Hengduan
Mountains of southwestern China. Three days after the wildfire, the first debris flow was triggered in
branch No. 3, followed by 2 other debris flows that same year. In August 2015, another debris flow
occurred in branches No. 1, No. 2 and No. 3, respectively. Rainfall data from three nearby rain gauges
and rainfall totals speculated from debris flow volume suggest the three debris flows in 2014 were
generated by isolated convective rainfall. Later, we found that varied rainfall thresholds existed among
the branches and that these thresholds might be related to the geological and geomorphic characteristics.
The results show that 1) the thresholds of post-fire debris flows tend to increase as time passes; 2)
post-fire debris flows in the Reneyong Valley occur with high frequency not only because of the loss of
the natural canopy, the occurrences of an ash layer and dry ravels and an increase in soil water
repellency but also because of the geology, drainage area, channel gradient and regional arid climate,
which may not be affected by wildfire; and 3) the varied rainfall thresholds among the different
branches are dependent on the drainage area, as entrainment is controlled by the magnitude of
discharge.
*Key words:* wildfire; debris flows; multiple factors; rainfall threshold
**1. Introduction**

Wildfires can quickly destroy vegetation and change the features of mountainous areas, resulting

in a high erosion rate (Conedera et al., 2003; Lane et al., 2006; Nyman et al., 2015; Orem and Pelletier,



2016) through debris flows, debris floods, debris slides, etc. The likelihood of post-fire debris flows
increases in proportion with burn severity (Jordan, 2016). Post-fire debris flows are caused mainly by
the runoff-triggered entrainment of hillslope material (Cannon et al., 2001; Santi et al., 2008; Kean et
al., 2011, Parise and Cannon, 2012) and infiltration-triggered meter-scale shallow landslides (Wondzell
and King, 2003; Cannon and Gartner, 2005; Parise and Cannon, 2012). Statistics show that the majority
of post-fire debris flows are triggered by surface water runoff (Gartner, 2005; Parise and Cannon, 2012;
DeGraff et al., 2015), especially in the first 1~2 summers after a wildfire, when large quantities of ash
from burned vegetation and unaggregated fine-grained dry ravel are susceptible to overland runoff or
debris flows. As time passes, the underground roots decay, and more infiltration-triggered meter-scale
shallow landslides emerge and transform into debris flows(Jordan, 2016; DeGraff et al., 2015).
Surface water runoff can be generated when the rainfall intensity is greater than the infiltration
rate. Wildfire can quickly destroy the vegetation in mountainous areas, making it possible for rainwater
to directly reach the earth rather than being intercepted by the canopy (Robichaud, 2000; Wondzell and
King, 2003; Larsen et al., 2009; Stoof et al., 2012). The sealing of surface soil pores by ash remnants
and other unaggregated fine soil particles can reduce the infiltration capacity of the soil (Larsen et al.,
2009; Woods and Balfour, 2010). In addition, soil water repellence increases after a fire so that the soil
resists wetting for a time and its soil hydraulic ability declines (Doerr and Thomas, 2000; MacDonald
and Huffman, 2004; Doerr et. al., 2006; Nyman et al., 2010). The hydraulic conductivity related to soil
sealing, soil water repellency and other hydrological properties(MacDonald and Huffman, 2004;
Moody et al.(2016). Moody et al.(2016)suggest that soil hydraulic conductivity is unchanged and
remains equal to the values for soil unaffected by fire for low burn severity and exponentially decreases
with burn severity when it is high. In short, a change of the surface soil properties caused by wildfire
can significantly decrease the soil hydraulic conductivity and induce greater surface water runoff for a
given amount of rainfall (Robichaud et al., 2000; Onda et al., 2008; Moody and Ebel, 2012; Moody and
Ebel, 2014). In addition, the rainfall threshold for debris flows can greatly decline after a wildfire
(Conedera et al., 2003; Cannon et al., 2008; Moody and Ebel, 2012; Staley et al., 2013).
The existing research on post-fire debris flows focuses more on western America (Robichaud et al.,
2000;Cannon et al., 2008; Larsen et al., 2009; Kean et al., 2011; Moody et al., 2016), followed by
southeastern Australia (Lane et al., 2006; Nyman et al, 2010; Smith et al., 2010); British Columbia,
Canada (VanDine et al., 2005; Jordan, 2016);and Switzerland (Conedera et al., 2003). In western China,





certain debris flows affected by wildfirehave been reported in Yangfan (Yunan Province) in the 1970s,
Jiarongka (Sichuan Province) in 2015 and Reneyong (Sichuan Province)in 2014 and 2015;however, no
detailed research has previously been conducted. This manuscript aims to:1) document the post-fire
debris flows in western China; 2) explore the effects of the inherent climatic, geologic and geomorphic
characteristics on post-fire debris flows; and 3) determine the reasons for the variations in rainfall
threshold among debris flows.

## 2. Study area

### 2.1 Natural setting

The Reneyong Valley, located in Xiangcheng County in the central Hengduan Mountains of
western China, covers an area of 24.28 km$^2$ with an outlet to the DingquRiver (which flows into the
Jinsha River, upstream of the Yangtze River) at 29°08′N, 99°33′E (Fig. 1). This catchment has a nearly
equilateral triangular shape and is surrounded by high mountains reaching 4222 m a.s.l. at the
northernmost location and 2855 m a.s.l. at the westernmost location on a fork of the Dingqu River.
There are 9 branches (No. 1~No.9) in this watershed (Fig. 1), and the width of the channels varies
between 2 and 30 m. In general, the branches have V-shaped channels and the main channel is
U-shaped. The geographic information for the three branches where debris flows occur is listed in
Table 1.Branches No. 1, No. 2 and No. 3 are in the southern part of the catchment and have an
elongated shape and a southeast-northwest orientation. Branches No. 1 and No. 2 have a similar
channel gradient, but branch No. 3 is much gentler. The change in gradient along the stream is similar,
with the steepest gradient in the central part and a relatively gentle gradient in the upper and lower
portions.
The continental monsoon plateau climate prevails in the study area, with rainfall concentrated
from June to September and plentiful sunshine. According to the statistics of rainfall data from the
Xiangcheng meteorological station (approximately 33 km to the southeast), the mean annual rainfall is
472.62 mm and the mean annual evaporation capacity of 2362 mm is 5.28 times the mean annual
rainfall, indicating that the study area is quite arid. Tall trees, shrubs and herbs cover the entire
watershed, and the majority of the trees are Pinus densata.
Faults tend to have a north-south orientation, and no single fault extends through the
watershed(Fig. 2). As in the Three Parallel Rivers area, the Neotectonic movement is strong with the





uplift of the Tibetan plateau; however, historically, earthquakes in Xiangcheng was reported to be lower
than Ms. 6.0, and the strongest recorded earthquake (Ms. 5.3) occurred on 5June1974 in Dongjun
village, 27 km northeast of this catchment. The bed lithology is soft rock and is divided into 4 units(Fig.
2) nearly parallel to the fault: black slate and sandstone of the Triassic system upstream, sandy slate and
some limestone of the Triassic system in the middle stream, black slate and some limestone of the
Triassic system on both sides downstream and alluvial deposits of the Quaternary system on the
channel bed and in the accumulation fan downstream.
**2.2 Debris flow cases**
Ancient debris flow deposits exist in the accumulation fan, indicating historical debris flows.
Interviews with local citizens indicated that no debris flows occurred in the 100 years before 2014,
while at least 4 debris flows have occurred since the wildfire in June 2014 (Table 2).
The town of Zhengdou is on the accumulation fan of the Reneyong Valley. On 8 June2014, the
local administrators were holding a seminar on reconstruction after the wildfire and were warned of
debris flows by a patrolman (Jiuli) who was responsible for geological hazards after he had found no
water flows in the channel. This first debris flow is identified as DF1 in this paper. After that debris
flow, the Sichuan Institute of Geological Engineering Investigation was appointed to conduct a field
survey. On 30 June, another debris flow occurred that is identified as DF2. On the night of 10July2014,
when we were staying at the local elementary school, we heard the noise of a debris flow collision and
then witnessed the debris flows in the downstream (this event is identified as DF3). On 24August2015,
a storm was predicted by the weather report, and the local geologic hazards administrator issued a
warning. Before the debris flows reached the downstream area, the patrolman (Jiuli) again found no
water flows in the channel and warned the local people to escape. This event is identified as DF4.
Fortunately, the 4 debris flows were reported before they reached the village, and although the debris
flows destroyed houses (Fig. 3), roads (Fig. 4) and farmland, leading to an economic loss of 18 million
Yuan, no people were killed.
In fact, as stated by a local citizen, in 2014, there were other debris flows rushing out from branch
No. 1, carrying dozens or hundreds of cubic meters of sediment downstream and cutting off the
Xiangcheng-Derong road. As the debris flows did not hit residential areas or destroy other facilities, the
exact time of the debris flows remains unclear. In general, debris flows after the wildfire in Reneyong





Valley is of high frequency.
**3. Methods**
**3.1 Meteorological data**
In western China, most rain gauges are located in the valleys, where there are more residents and
the basic facilities are better, while few exist in the upper areas where debris flows begin. The study site
is in the central Hengduan Mountains, and the nearest three rain gauges, at Zhengdou, Adu and Reda
near the study area, are applied(Fig. 2, Table 3). The firs train gauge, Zhengdou, is on the deposition
fan of the Reneyong Valley; the second, Reda, is in another valley on the other side of the southeastern
crest; and the third, Adu, is in the same valley as Reda on the other side of the northeastern crest. The
three rain gauges form a triangle around the study area, monitoring rainfall sources from several sides.
Other information about the three rain gauges is listed in Table 3.
As rainfall arriving at the initiation area can be transformed in different ways, rainfall data from
the nearer rain gauge could be more important for determining the average rainfall process. Indeed, the
reciprocal-distance-squared method can be used to deduce the average rainfall process in the initiation
area as follows(Chow et al., 1988; Chen et al., 2012):
$$P = \sum_{i=1}^{3} \omega_i P_i \qquad\qquad (1)$$
where $P_i$ is the rainfall record from the rain gauges; $i$ =1, 2, or 3 represents the Zhengdou, Reda and
Adu rain gauges, respectively; and $\omega_i$ is the weighing factor corresponding to $P_i$. The weighting
factor can be expressed by $\omega_i = d_i^{-2} / \sum_{i=1}^{3} d_i^{-2}$, where $d_i$ is the distance from rain gauge $i$ to the
debris flow initiation area.
**3.2 Field survey**
After the debris flows, we conducted a field survey to evaluate the impact of the wildfire and
investigated the initiation process and the magnitude of the debris flows to propose debris flow
alleviation strategies. After DF1, we conducted the first field survey and personally witnessed DF2
moving downstream when we were living in the local elementary school. After DF4, we conducted a
second field survey to investigate the debris flow imitation process and examine the impact of debris



flows on the check dams to evaluate whether additional work was required to prevent future debris
flows.

(1) Detecting the scope of the wildfire

We interviewed the local citizens and were informed that the fire was accidentally set by workers

who were building an iron tower for an electrical transmission line at 18:00 on 1 June 2014. After the
fire, fire fighters, armed police and local citizens gathered to fight the fire and it was extinguished at
10:00 on 5 June.

After the wildfire, the forest administrators measured the scope of the wildfire. They walked along

its boundary and marked the scope on a contour map (with a scale of 1:100000). According to this map,
an area of 5.4 km$^2$ in the catchment was affected by the wildfire (Fig. 1), accounting for 22.2% of the
entire watershed. In detail, branches No. 1, No. 2 and No. 3 were within the scope. The majority of the
trees are Pinus densata, under which are shrubs and herbs, a good place for yaks and sheep to graze
(Fig. 5).

(2) Sediment investigation

Our field survey was conducted along the channel, and a laser range finder was applied to gather

measurements. We measured the bank failure and the high erosive deposits along the two sides of the
channel, the bank-failure induced soil slide, and the scope and amount of spoil along the
Xiangcheng-Derong road. We marked these on a contour map and recorded them in a notebook,
respectively. We dug six troughs, 1.5m in length, 0.5 m in width and 0.3~0.6 m in depth (Fig. 6), on the
slope where the wildfire burned to detect the depth of the ashes and the variety and extent of roots
destroyed by the wildfire. We also collected soils from the troughs to measure the particle size
distribution and the natural water content. In addition, a borehole was used to measure the depth of the
debris flow deposits and the loose gravel deposits underneath.

(3) Measurement of debris flow deposits

The volume of debris flows can be used to evaluate the magnitude, which can be found by

measuring the sporadic deposit division. For each deposit division, we used a laser range finder to
measure the scope and average depth to calculate its volume. The precision of the volume was more
dependent on the measurement of the average depth, which can reach 100 m$^3$, thus the volume of each
division larger than 50m$^3$ would be recorded as 100 m$^3$, otherwise, it would be not included.

The majority of DF4 is deposited behind two newly built check dams and only a few portions




reached downstream by passing through cracks on the check dams. We measured and marked the
boundaries of the deposits on the contour map that was completed during the first field survey before
the check dams were built. We measured the height of the check dams above the deposits and obtained
the depth buried by the deposits, which is the greatest depth of the deposit. We divided the largest
deposit depth into several parts and calculated the volume as follows:
$$V = \sum_{i=1}^{n} h_i A_i \qquad (2)$$

where $V$ is the volume of the deposits; $h_i$ is the height of each part of the deposits; $A_i$ is the area of
the horizontal area; $i$ =1, 2, …… , and n represents the parts that were divided. Normally, we set $h_i$
=1m, which means that the deposits from the toe of the dam to the end of the deposit after the dam
were divided into n parts and that each part had a height of 1m. $A_i$ is the area circled by the axis of the
dam and the corresponding contour line and can be obtained using a 1:500 contour map.
In the deposit zone, we measured particle size and lithology. We placed a ruler of 50 m on the
surface of the deposits randomly. We measured particle size and recorded the lithology of the stones at
a 1-m interval along the ruler (Fig. 7). For particles larger than 60 mm, the diameter and lithology were
recorded, otherwise, only the lithology was recorded. Deposits smaller than 60 mm were collected to
complete particle size distribution tests in the laboratory.

## 4. Analysis and results

### 4.1 Recorded rainfall process

Before the debris flows daily rainfall data were collected from Zhengdou, Reda and Adu, and the
reciprocal-distance-squared method was applied to obtain the average rainfall(Table 3). Table 2 shows
that in 2014, there was only occasional drizzle in the preceding days and the 3-day accumulated rainfall
was only a few millimeters except in the case of DF3, when it was 14.84 mm. In 2015, it sprinkled for
several days, and the 3-day accumulated rainfall before DF4reached nearly 40 mm (daily rainfall data
for the day before DF4 is missing because of instrument error, and rainfall data from the Xiangcheng
meteorological station, 33 km to the southeast, were used). In the year that the wildfire occurred, the
3-day accumulated rainfall for the 3 debris flows varied greatly, which suggests that post-fire debris
flows were not correlated with the antecedent rainfall; however, the antecedent rainfall significantly



increased in the second summer.
Hourly rainfall data before and after the debris flows were collected from Zhengdou, Reda and
Adu, which were applied to determine the average rainfall process by the reciprocal-distance-squared
method. The rainfall processes before and after the four debris flows are depicted in Fig. 8, which
shows there was short-term low-intensity rainfall before the three debris flows in 2014. The rain gauge
worked well, as local administrators discussing reconstruction after the wildfire recalled that only
occasional drizzle had occurred prior toDF1. When we were living in the local elementary school we
witnessed only sprinkles when DF3 occurred. However, it seems impossible for a rainfall intensity of
1mm/h or less to generate sufficient surface runoff and the subsequent debris flows. Instead, local
convective rainfall in the mountains could be the trigger which cannot be recorded by the nearby rain
gauges.
On 24 August2015, the storm began at approximately 19:00, and the average rainfall intensity
reached 26.39 mm/h. In the outlet of the main channel, the rainfall intensity reached 38.5 mm/h and
then declined to less than 5 mm/h. DF4 arrived in the downstream area at approximately 19:43, and if
we deduct the time needed for it to move from the initiation area to the outlet, we find that the debris
flows were likely initiated shortly after the rainfall began.
**4.2 Debris flow initiation process**
According to the location of the debris flow deposits and the residual scar left by debris flow
erosion, all the debris flows originated in the fire-affected area, with DF1, DF2 and DF3 deriving from
branch No. 3 and DF4from branches No. 1, No. 2 and No. 3 and some smaller neighboring catchments
on 24August2015. However, as some catchments' drainage is too small to depict accurately in Fig. 1,
this paper considers only branches No. 1, No. 2 and No. 3.
(1) Debris flows in 2014
In 2014, the debris flows were triggered in the upstream area of branch No. 3. Upstream, trunks
were surrounded by charcoal and ashes, and only a few trunks toppled over. Shrubs, herbs and litter
were consumed by the wildfire, and the slope surface was covered by ashes, but the underground root
system survived and the affected soil was concentrated in only a few centimeters. Unaggregated dry
ravel is widely distributed, and dry ravel remnants mixed with ashes were found to have flowed along
the slope, indicating that the entrainment of the surface runoff should be the origin of the debris flows.



The channel is only 0.5~0.8 m wideand0.4~0.6 m deep on a steep slope with exposed stones and no
debris flow deposits (Fig. 9). Items above the channel fell parallel to the channel, suggesting that the
debris flows submerged the entire channel and that the discharge was still smaller than 1 m³/s. Erosion
caused by the debris flows is limited by the small discharge, and the transported soil particles were
limited in the smaller debris flows. In the middle stream and downstream areas of branch No. 3, the
channel gradient decreases, while the slope of both sides increases to 35~45°, forming a narrow
V-shaped gully. The moving debris flows entrained the bed sediment and scored the base of the banks,
leading to bank failure on both sides. The intensive scars of landslides with no vegetation can be found
on both sides along the channel. These shallow landslides were meter-scale, with a volume ranging
from tens of cubic meters to thousands of cubic meters. At the beginning, where the channel is quite
narrow, it can be blocked by landslide deposits of a small magnitude; as more sediment is deposited in
the channel, the broadened channel can be partly blocked by a small landslide and entirely blocked by a
large one (Fig. 10). In addition, the burned trunks can favor channel blocking. Channel blocking
alleviates the debris flow process, and the outburst debris flows have a significantly larger discharge
(Cui et al., 2013; Zhu, 2013).
(2) Debris flows in 2015
The debris flow initiated in branch No. 2 (Fig. 11) is similar to that initiated in branch No. 3, while
that initiated in branch No. 1 (Fig. 12) is slightly different. Shrubs, herbs and the lower parts of the
trees were partly consumed by the wildfire. In the second summer after the wildfire, the trees were
again covered by green crowns. Branch No. 1 can be divided into three parts according to the channel
gradient, with the steepest gradient (32°) in the middle part, where the debris flow was initiated, and a
gentle gradient in the source area. Two smaller gullies converge at the debris flow initiation zone,
forming a platform between them. During the storm, the surface water runoff in the source area
entrained sediment and formed a debris flood. Following the debris flow initiation zone, which is quite
steep, the debris flood from the two gullies had a higher erosion ability; it scored both sides of the
platform and triggered bank failure, followed by the retrogressive meter-scale landslide failure of the
platform. The debris flood mixed with the detached bank slope and formed debris flows; meanwhile,
the 9.2 mm of accumulative rainfall over the previous three days endowed the surface layer with
relatively higher water content, and the retrogressive landslide failure caused it to slide and liquefy into
debris flows. In the lower stream, a debris flow moving over wet sediment can greatly increase



sediment entrainment and significantly amplify the magnitude of the flow (Iverson et al., 2011; McCoy
et al., 2012).

**4.3 Debris flow deposits**

The majority of DF1 deposited in the wide section of the channel downstream of the fork with
branch No. 3. Some of it jumped the channel banks and destroyed houses, and the remaining rushed
into the Dingqu River though it did not block the river (Fig. 13a). DF2 traced the previous path, leaving
a slight depth of debris covering the DF1 deposits and striking our borehole instrument (Fig. 13b). The
volume of DF2 is much smaller than that of DF1. DF3 continually traced these deposits and left
considerable deposits in the wide section. DF3 also jumped the river banks and buried some parts of the
road in the residential area and the remaining partially blocked the Dingqu River (Fig. 13c). As two
check dams were completed, the deposits of DF4are much different. The majority of DF4 was
intercepted by the two check dams except a portion in the mainstream. Debris reached the top of check
dam No. 1 (with a height of 4 m) and only the upper 6 m of check dam No. 2 was above the deposits
(Fig. 13d), leaving the lower 12 m buried by the deposits.
Based on field measurements and indoor calculations, we determined the volume of the four
debris flows (Table 2). There are large differences among the volumes of the debris flows, of which, the
volume of DF4 is the largest, reaching 154,500m$^3$, followed by DF1at 86,200m$^3$, DF3at 3,2300 m$^3$ and
DF2at 5,100m$^3$. Although the volumes of DF2 and DF3are smaller than that of DF1, they still arrived
downstream and were dangerous, as DF1 had paved a path, and the friction of the stream had decreased
significantly. Statistics show the deposits are angular, and the majority of them are sandy slate
(90.48%), followed by slate (7.14%) and limestone (2.38%), which suggests that the debris flows
originated from branches Nos. 1~3, where sandy slate dominates, and this suggestion is consistent with
our field survey.

**5. Discussion**

Debris flows do not occur in all fire-affected watersheds; instead, the response to rainfall could be
debris flows (in a proportion of 40%) (Cannon, 2001; Nyman et al., 2010; Kean et al, 2011), flash
floods (Cannon, 2001; Kean et al. 2011)or no response (Cannon, 2001; Smith et al. 2010). The debris
flows in the Reneyong Valley are unusual because the debris flows are of high frequency; in addition,
although three debris flows occurred in branch No. 3 in 2014, branches No. 1 and No. 2 had no



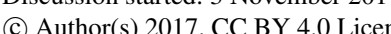


response to rainfall even though they had steeper gradients and debris flows are more likely to occur in
the first year following wildfire.

These facts lead us to believe that the likelihood of debris flows is correlated with the impact of

wildfire but also with the geology, drainage area, channel gradient, and regional climate, which are not
affected by wildfire. In the later discussion, we attempt to discuss these factors to resolve the doubt we
have encountered regarding the post-fire debris flows in the Reneyong Valley.
**5.1 Rainfall threshold**

The 4 post-wildfire debris flows in the Reneyong Valley were generated by surface water runoff.

The rainfall intensity recorded by the downstream rain gauge was 1 mm/h or less and the duration was
quite short, which is in line with the memory of the local citizen in the downstream area who witnessed
the flow, suggesting the rain gauge was working well; however, it seems impossible for a low-intensity
short-duration rainfall to generate surface water runoff, let alone entrain sediment and trigger debris
flows. In the Hengduan Mountains, isolated convective rainfall is common and has been found to be an
important trigger of debris flows in this area (Tang et al., 2011; Ni et al., 2014); in addition, rainfall
intensity has also been found to increase with elevation in the Jinsha Basin(Tan et al., 1994). Here, we
speculate that the three post-wildfire debris flows in 2014 were triggered by isolated convective rainfall
and that the rain gauges down slope can definitely record the triggering rainfall.

Observations of debris flows in the Jiangjia Valley show that higher intensity rain can generate

debris flows of larger magnitude, and an exponential increasing model was built (Zhuang et al., 2009)
that might model the process of rainfall amplifying the activity of debris, resulting in more soil slides
and of greater magnitude (Dai and Lee., 2001; Guo et al., 2013). In addition, sediment wetted or
saturated by rainfall is more susceptible to entrainment by debris flows (Iverson,2011;McCoy et al.,
2012). Similar research can be found in post-wildfire research, just as rainfall totals have been applied
in the magnitude prediction model, other factors, including drainage area and burned areas of high and
moderate severity, have been incorporated (Gartner et al., 2008; Cannot et al., 2010).

In a given catchment, we attempted to use the volume of the debris flows to deduce the possible

triggering rainfall as follows: the volume of DF4 is much larger, suggesting the highest triggering
rainfall, followed by DF1, DF3 and DF2, respectively. In a word, low rainfall in 2014 did trigger three
debris flows in branch No. 3, while none occurred in branches No. 1 and No. 2. Greater rainfall in 2015





generated debris flows in the three branches, which indicates that the rainfall threshold for post-wildfire
debris flows in the Reneyong Valley was quite low during the first summer after wildfire and that it
increased as time passed. In addition, debris flows in branches No. 1 and No. 2 had a higher rainfall
threshold compared to that of branch No. 3.
**5.2 Regional climate**
Soil water repellency could be of high significance in reducing soil hydraulic conductivity and
amplifying surface water runoff immediately after a wildfire(MacDonald and Huffman, 2004; Doerr et.
al., 2006; Moody et al., 2013). It may also inhibit the soil rewetting process (Doerr et al.,2000), which
may require days to weeks and will be quite small or nonexistent in the second summer after a fire
(MacDonald and Huffman, 2004; Larsen et al., 2009).According to our trough test, soil in the burned
area was covered by a centimeter of ashes and the surface layer affected by wildfire was concentrated
in only a few millimeters(Fig.6), and the soil water repellency of the surface layer should have been
limited, which might have played a key role in triggering DF1 and might have had no effect on DF4.
The surface soil could have low water content as a result of the long-duration arid climate, so that
the surface layer could also have low hydraulic conductivity (Moody and Ebel, 2012; Sheridan et al,
2016). This low conductivity may be responsible for the quite low rainfall threshold for the later debris
flows, as rainfall infiltration into the soil is limited and surface runoff can easily occur. Indeed, if a
hyper-dry condition is reached, no rain can infiltrate into the soil (Moody and Ebel, 2012). The effect of
a long-duration arid climate on soil water content could be meter-scale, while the depth of rainfall
infiltration is limited to the surface and the time for the soil to recover aridity could be only days.
Aridity should be a key theme because the drought ravel in steep arid catchments has been identified as
an important source of runoff-triggered debris flows (Kean et al., 2011, 2013; Staley et al., 2014; Noske
et al., 2016).
It is highly difficult for vegetation to recover in newly generated landslide scars in an arid climate,
and the uncovered loose sediment can be much more easily entrained by debris flows without the
binding effect of roots (Ziemer,1981; Gyssels et al., 2005). In general, the increase in soil water
repellency and decline in hydraulic conductivity induced by wildfire should be transient and the effect
of an arid climate on erosion could be perennial, which has been verified in the non-fire-affected area
(Carretier et al.,2013).



### 5.3 Geology and soil properties


Sandy slate from the Triassic system dominated the three branches, accompanied by small
amounts of limestone. Sandy slate is soft and susceptible to the weathering process, resulting in a deep
mantle of soil covering the bedrock with a high fine-particle content and more boulders of limestone.
Based on the field survey of the successive landslide scars along the branches, the sediment is fine
grained, arid and loose, characteristics that make it vulnerable to debris flow entrainment. As the
weathered eluvium is abundant, the channel is charged with sediment, resulting in a high frequency of
debris flows (Bovis and Jakob, 1999; Jakob et al., 2005). In the branches, the underlying bedrock can
hardly be found, and bedrock in some sections of the main channel is uncovered, suggesting that the
downward erosion of debris flows is an important process in the steep branches and that the successive
bank failures are generated by debris flow bulking(Hungr et al., 2005; Gabet and Bookter, 2008; Zhu,
2013). These landslides can partly or wholly block the channel, and debris flows can be greatly
amplified after an outburst(Cui et al., 2013; Zhu, 2013).

### 5.4 Channel gradient and drainage area


The three catchments share a similar channel gradient, with the largest gradient in the central part
and smaller gradients in the upper and lower parts (Table 1). This configuration tends to produce a
greater surface runoff for a given rainfall process and to exert higher erosion ability in the middle area
with the largest gradient to produce debris flows of greater magnitude(Coe et al., 2008; McCoy et al.,
2012), as entrainment in steep terrain can increase rapidly with slope owing to both shearing stress and
transport capacity (Foster and Meyer, 1972; Stock and Dietrich, 2003; Hungr et al, 2005; Moody et al,
2013; Kean et al, 2013).
Drainage area, the scope of the area that rainfall can flow into, is a more dominant factor for the
magnitude of surface runoff. Of the three sub-catchments, branch No. 3 has the largest drainage area,
followed by branches No. 2 and No. 1, each of which has a drainage area smaller than 1 $km^2$. In 2014,
the debris flows originated solely in branch No. 3; however, in 2015, debris flows occurred in branches
No. 1~3 and some smaller sub-catchments where the rainfall intensity reached 38.5 mm/h. Here, we
hypothesize that the terrain is similar: a larger area tends to have greater surface water runoff, and the
likelihood of debris flow occurrence could be higher; thus, greater rainfall is required to trigger
post-fire debris flows in a relatively smaller area.



This principle should not be applied in all fire-affected areas, as the statistics developed in earlier
studies (Gartner, 2005; Cannon et al., 2010)suggest that post-fire debris flows can occur where the
drainage area is smaller than 25 km$^2$ and even where it is smaller than 5 km$^2$. The reasons may be that
the terrain of a smaller catchment is apt to be steep and the surface water runoff can have higher
erosion ability (Hungr et al, 2005; Moody et al, 2013), increasing the susceptibility to debris flow
occurrence; as the drainage area increases, the catchments tend to have wider channels and gentler
gradients(Stock and Dietrich, 2003), resulting in smaller-unit runoff discharge and lower erosion ability.
When the drainage area is larger than 25 km$^2$, the unfavorable effects of a wider channel and gentler
gradient on post-fire debris flows might surpass the favorable effect of wildfire, resulting in no
response to the wildfire.
**5.5 Human activity**
In addition to the wildfire set accidentally by people, the construction of the Xiangcheng-Derong
road is another important factor for the amplification of debris flows. The Xiangcheng-Derong road
crosses the port on the eastern border and stretches along the main channel from the outlet of channel
No. 5. Approximately 13.64 km is distributed in the Reneyong watershed, and abundant spoils were
produced when it was constructed. These spoils were deposited on the southern slope of the main
channel with a gradient slightly larger than the friction angle and only a few retaining walls; thus, some
of them have reached the main channel, resulting in a narrowing of the channel. Spoils can also be
found in some other branches. The spoils are composed of fine-grained particles and are only slightly
covered by vegetation because of the arid climate. Although spoils outside the main channel were not
affected by the wildfire, these spoils are still arid, with low hydraulic conductivity owing to long-term
drought(Moody and Ebel, 2012; Sheridan et al, 2016). Low rainfall intensity is required to trigger
surface water runoff and the consequent debris flows(Coe et al., 2008; Kean et al., 2011), which can
partly or wholly block the channel (Fig. 14). This narrowed channel can also be blocked by the large
trunks carried by debris flows.
At the narrowed channel section, the debris flows would have a greater depth, resulting in an
increase in the debris flow velocity and greatly enhancing the subsequent erosion ability. A large
amount of the erodible spoils was enrolled by the debris flows, significantly amplifying the magnitude
(Cui et al., 2013; Iverson and Ouyang, 2015). After DF1, abundant spoils were entrained by the debris



flows, and remnants of the deposited spoils can be found in a steep section 2~4 m high(Fig. 15). From
the outlet of branch No. 3, there were more than 10 narrowed channel sections; one would induce a
significant entrainment process that could amplify debris flows.
## 6. Conclusion
The existing research on post-wildfire debris flows focuses mainly on the decline in hydraulic
conductivity resulting from the increase in soil water repellency (Doerr and Thomas, 2000; MacDonald
and Huffman, 2004; Doerr et. al., 2006; Nyman et al., 2010), the process of soil sealing(Larsen et al.,
2009; Woods and Balfour, 2010), and the low rainfall intensity needed to produce surface water runoff
and trigger debris flows. In addition, the geologic and geomorphic characteristic of the catchment that
may not be affected by wildfire can still produce favorable effects for the magnitude and frequency of
debris flows as follows: 1) The arid climate can reduce the soil water content and hydraulic
conductivity, which can have a positive effect on debris flows, as soil water repellency will quickly
decrease after rainfall; in addition, the arid climate leads to slow vegetation recovery. 2) The deep
weathered remnant of sandy slate has high fine-particle content and high susceptibility to debris flow
entrainment; therefore, the watershed is charged with abundant sediment. 3) The "gentle-steep-gentle"
gradient can contribute to greater surface water runoff and the subsequent severe erosion process in the
steep area. 4) The downward erosion of debris flows in the steep branches generates successive bank
failure, which amplifies debris flows. 5) Statistics show that post-fire debris flows tend to occur in
catchments smaller than 5 km$^2$ (Cannon et al., 2010) and debris flows in smaller watersheds are apt to
be triggered by a higher rainfall threshold, such as for branches No. 1 and No. 2.

## Acknowledgements
This research was supported by the National Natural Science Foundation of China (grant Nos.
41661134012 and 41402283). We wish to acknowledge the editors of the Natural Hazards and Earth
System Science Editorial Office and the anonymous reviewers for their constructive comments, which
helped us improve the content and presentation of the manuscript.

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





Table 1. Geographic information for branches No.1~No.3

| No. | Drainage area (km$^2$) | Maximum basin relief (m) | Gradient (degrees) | Gradient (central part) (degrees) | Gradient (upper part) (degrees) |
|---|---|---|---|---|---|
| Branch No.1 | 0.37 | 544 | 27 | 32 | 18 |
| Branch No.2 | 0.73 | 755 | 26 | 35 | 18 |
| Branch No.3 | 2.30 | 836 | 15 | 20 | 14 |






Table 2. Basic information regarding the debris flows

| No. | Time | Location | Recorded 3-day antecedent precipitation(mm) | Debris flow volume (1000 m$^3$) |
|---|---|---|---|---|
| DF1 | 16:08, June 8, 2014 | No. 3 | 1.11 | 86.2 |
| DF2 | 16:00, June 30, 2014 | No. 3 | 14.84 | 5.1 |
| DF3 | 23:20, July 10, 2014 | No. 3 | 0.81 | 32.3 |
| DF4 | 19:43, August 24, 2015 | No.1, No.2 and No.3 | 39.97 | 154.5 |




Table 3. Basic information regardingthe three rain gauges

| No. | Name | Location | Elevation (m) | Distance (km) | Precision (mm) | $\omega_i$ |
|-----|------|----------|---------------|---------------|----------------|------------|
| 1 | Zhengdou | 29°08′N, 99°33′E | 2858 | 3.90 | 0.1 | 0.60 |
| 2 | Reda | 29°06′N, 99°38′E | 3363 | 5.56 | 0.1 | 0.30 |
| 3 | Adu | 29°11′N, 99°39′E | 2783 | 9.40 | 0.1 | 0.10 |







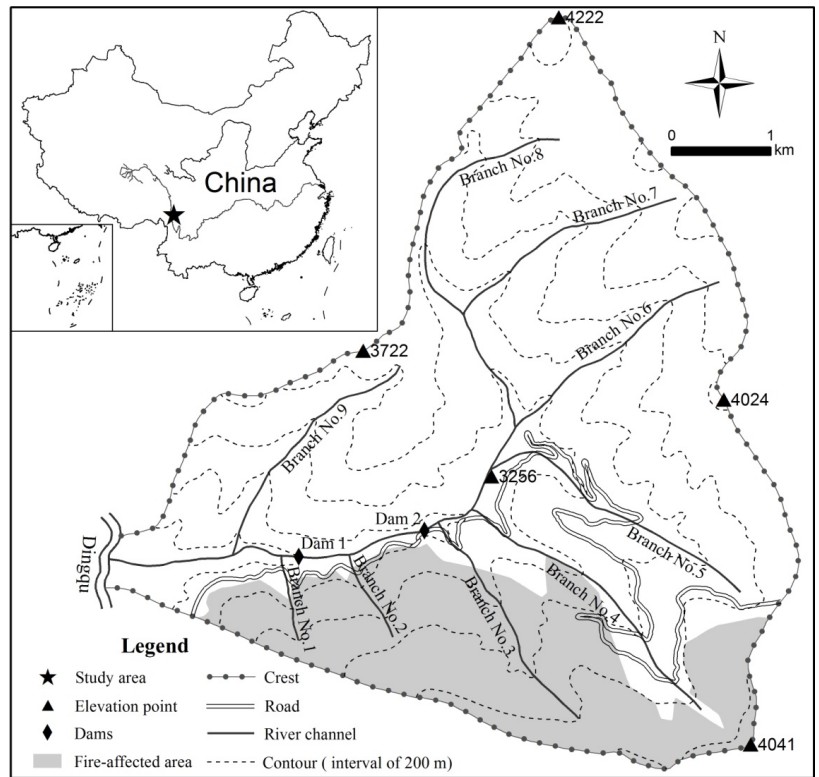

583      Figure. 1. Location of Reneyong Valley and related geographic features



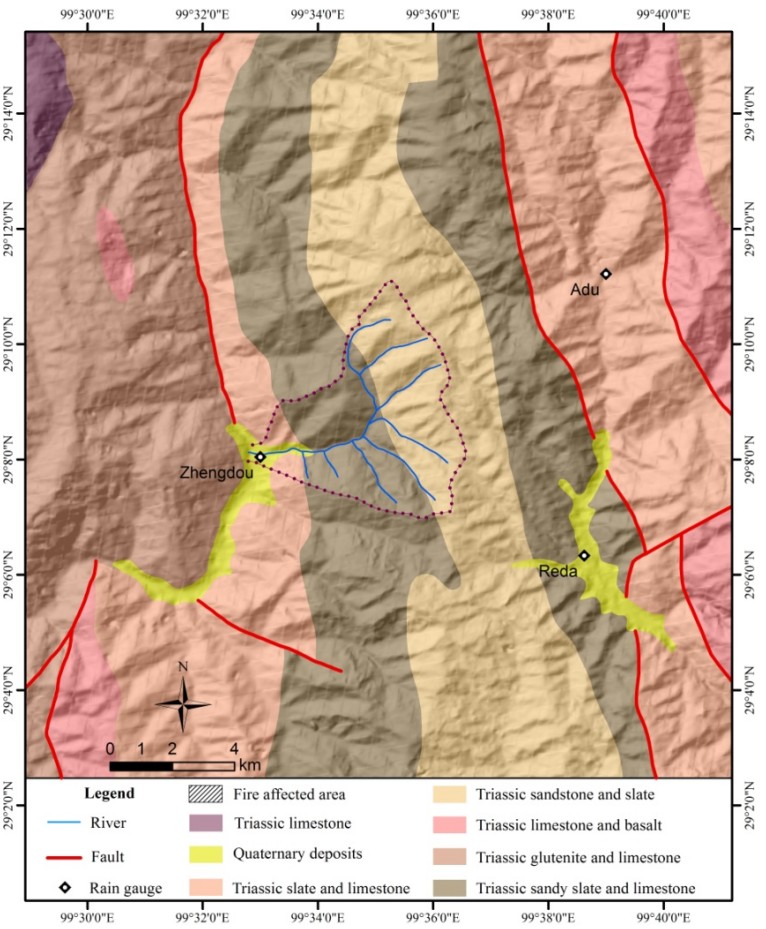


Figure 2. Simplified geologic map and distribution of applied rainfall gauges



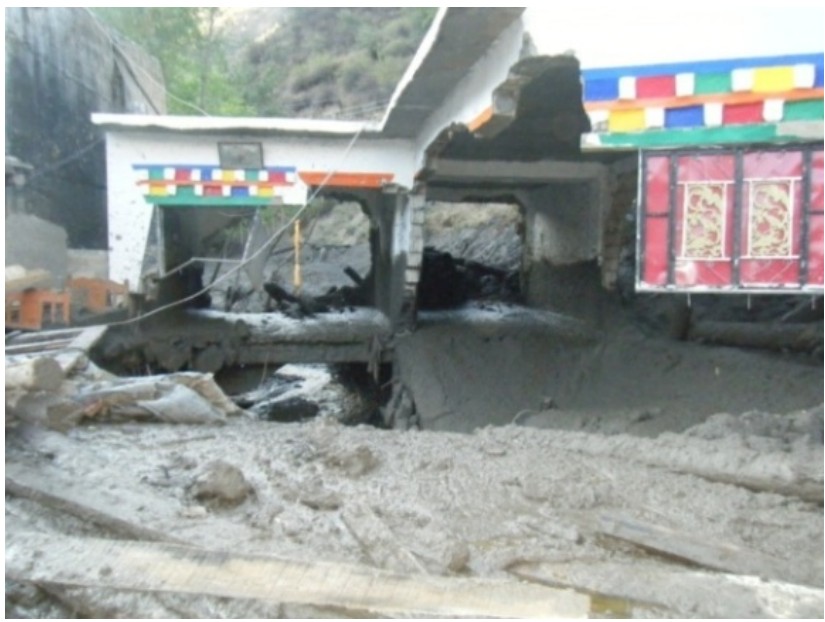


Figure 3. Houses destroyed by debris flows in the downstream





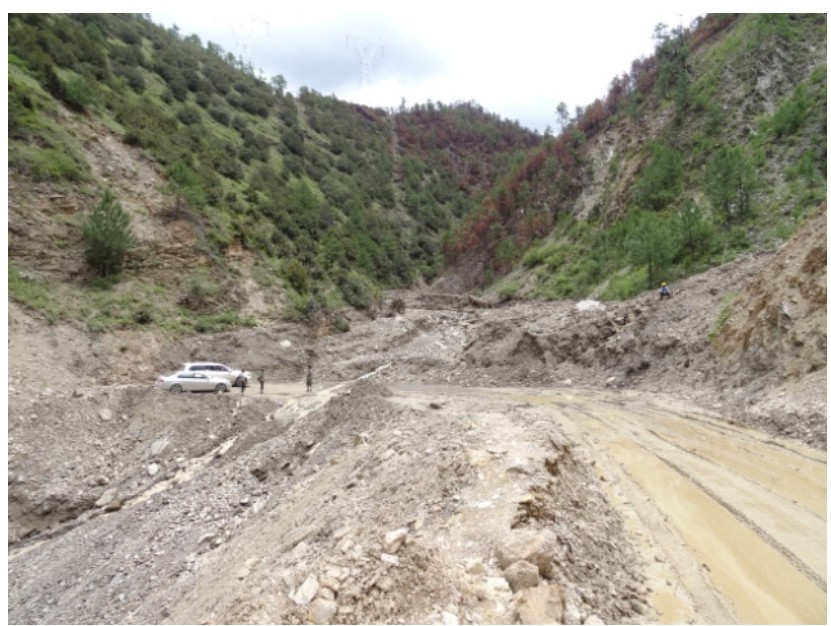


Figure 4. Road buried by debris flows from branch No. 3





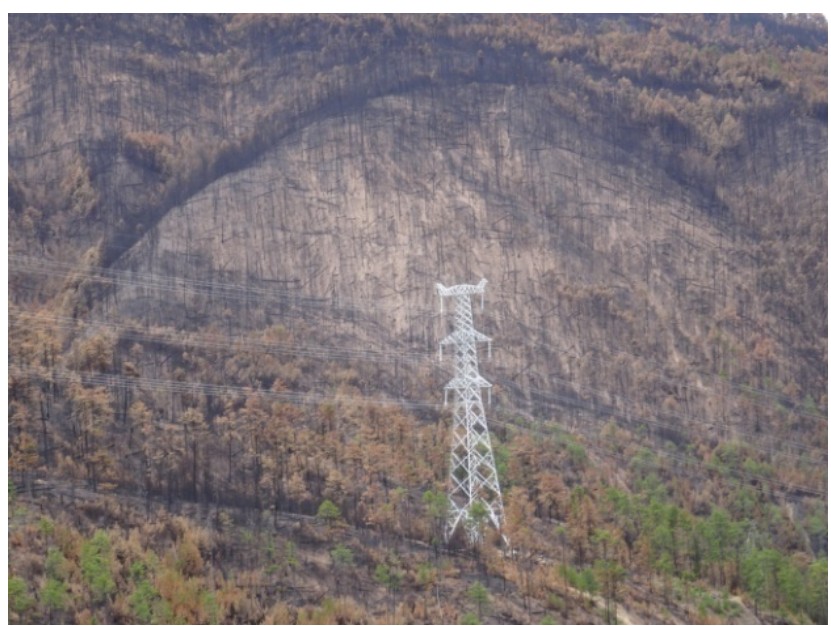


Figure 5. Overlooking the fire-affected area


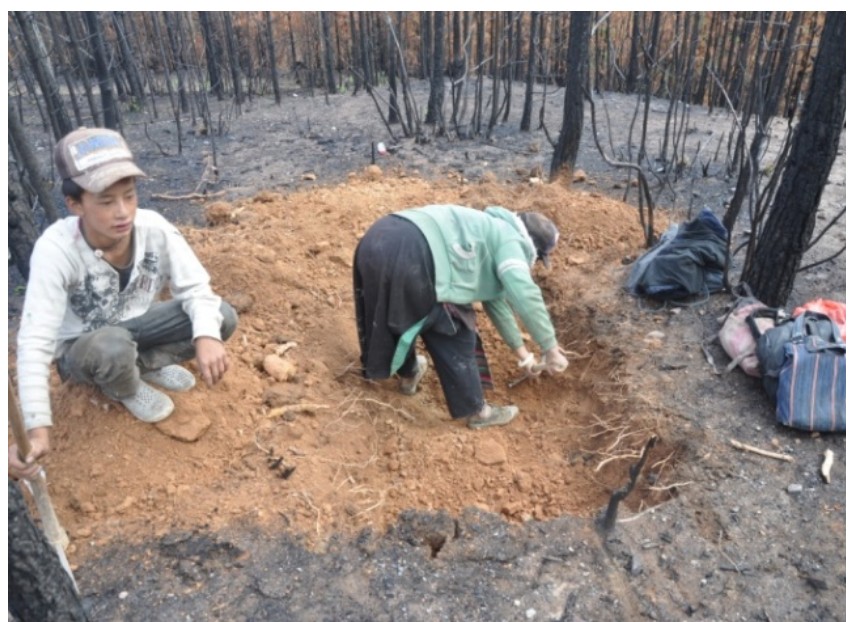


Figure 6. Trough test in the fire-affected area



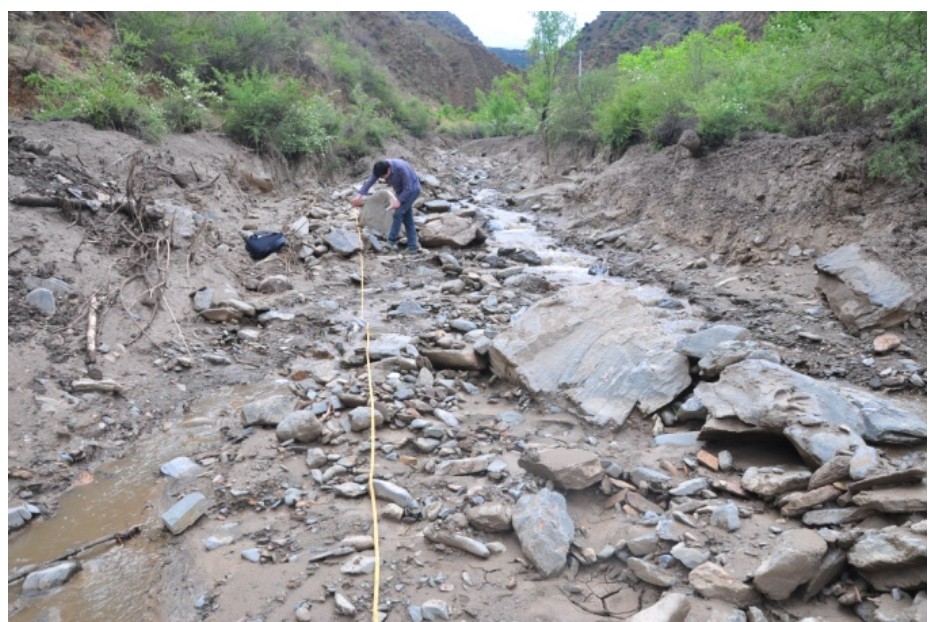


Figure 7. Particle size measurement of debris flow deposits(DF1)



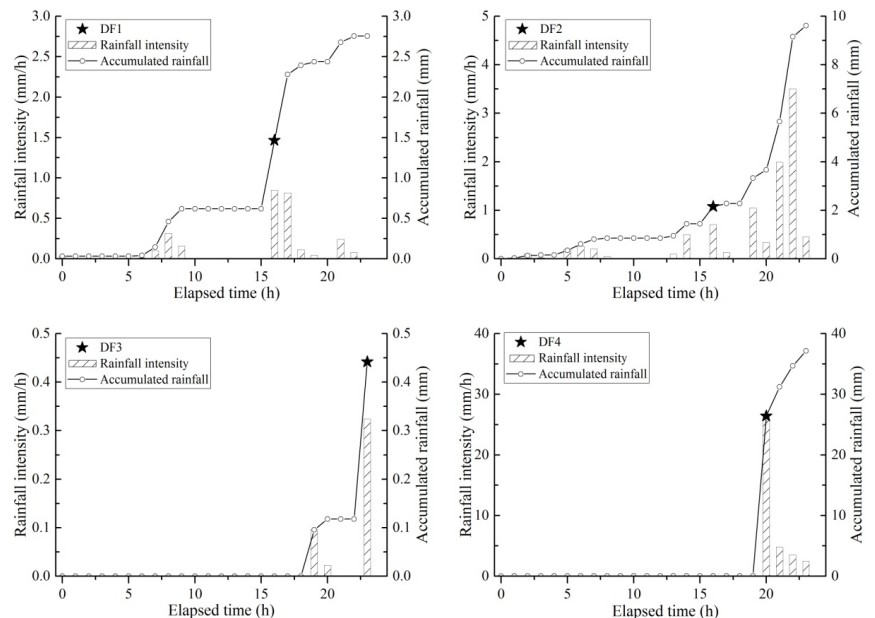


Figure 8. Recorded rainfall prior to the debris flows(rainfall data from theZhengdou, Reda and Adu rain

gauges were applied)




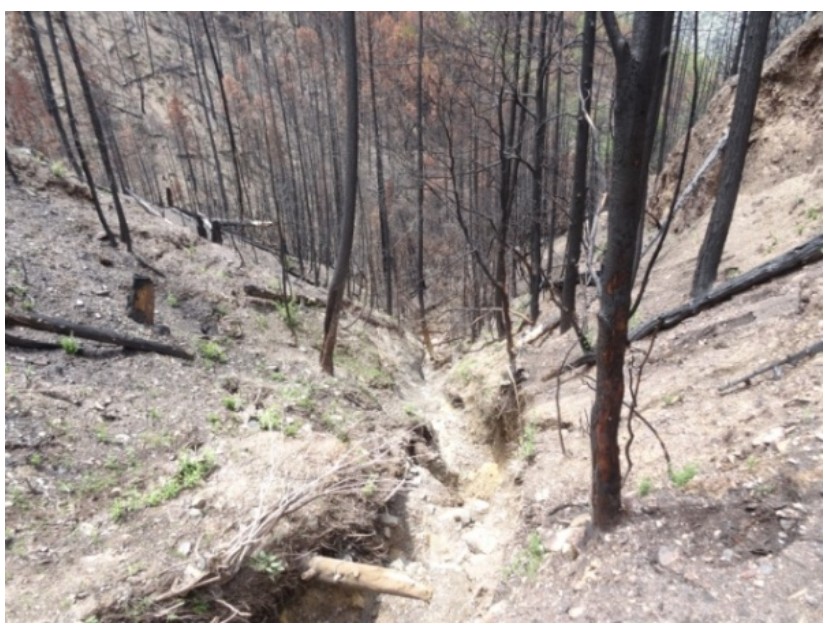


Figure 9. Upstream of branch No. 3 after DF1 (no debris flow remnants)






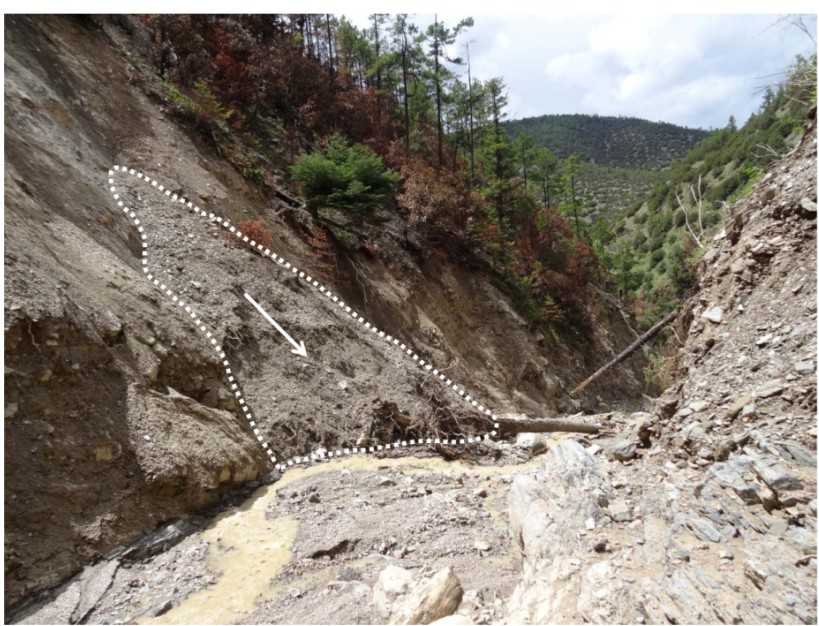

Figure 10. Middle stream of branch No. 3 after DF1

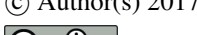


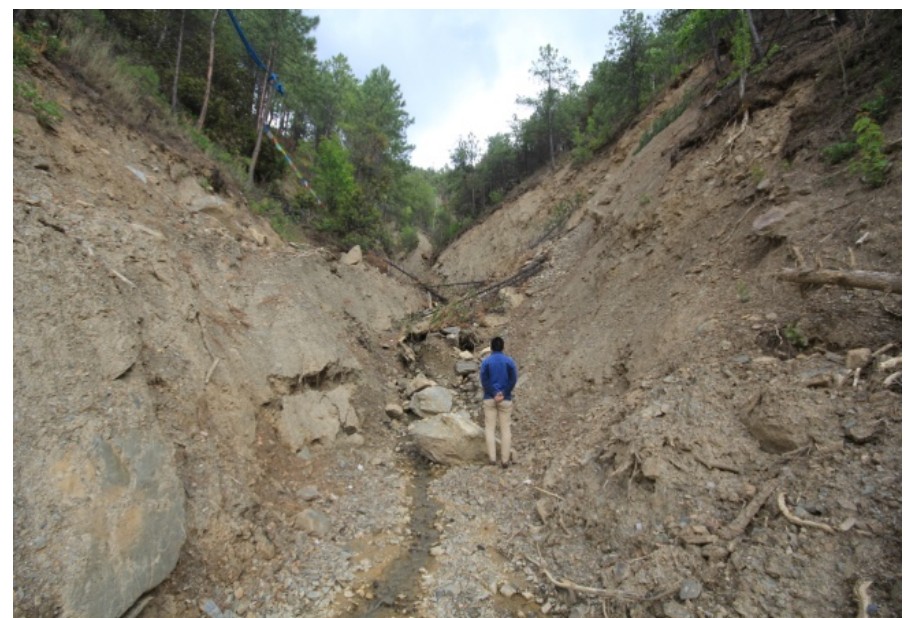


Figure 11. Branch No. 2 after DF4





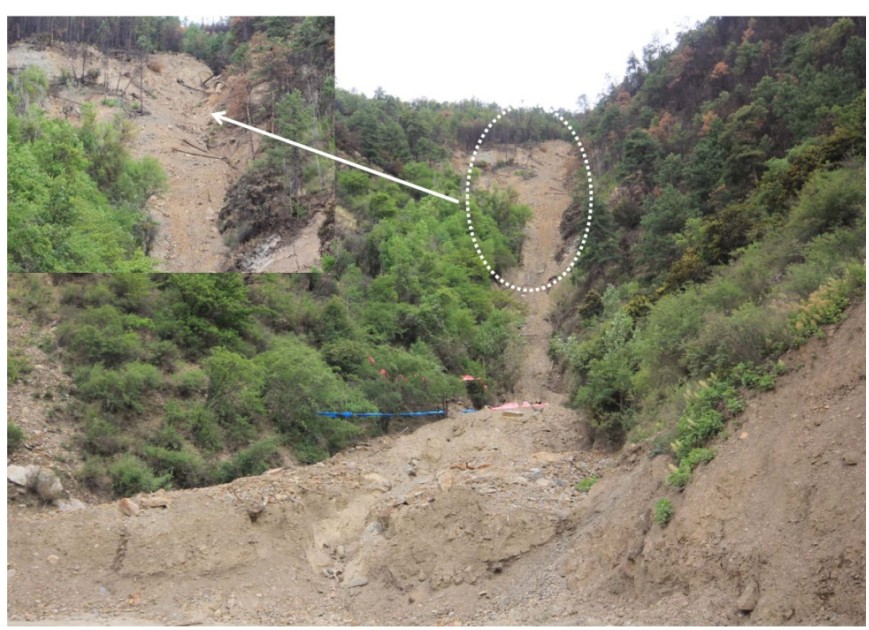


Figure 12. Branch No. 1 after DF4






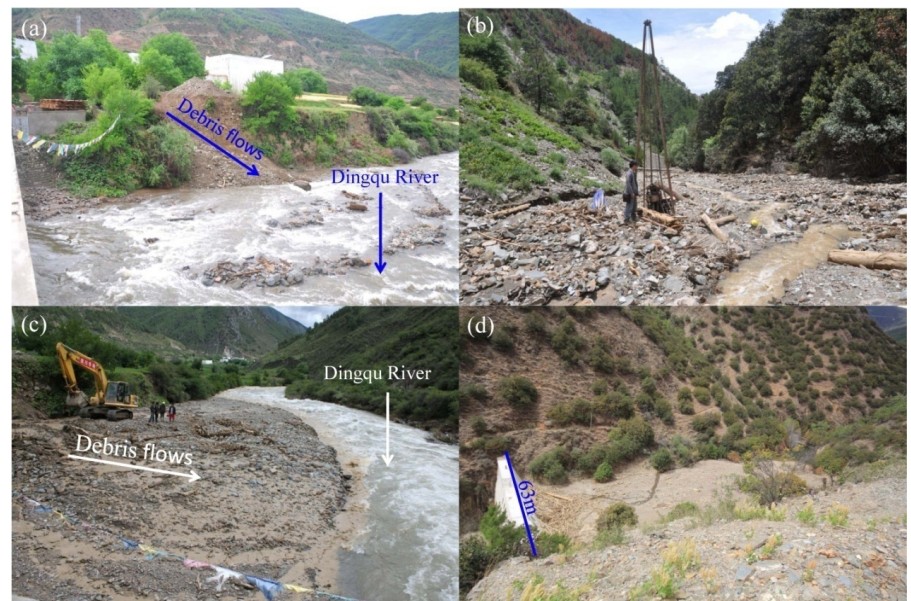


Figure 13. Debris flow deposits of (a) DF1 in the Dingqu River, (b) DF2 slightly striking the borehole

instrument, (c) DF3partly blocking the Dingqu River, and d) DF4 intercepted by check dam 2






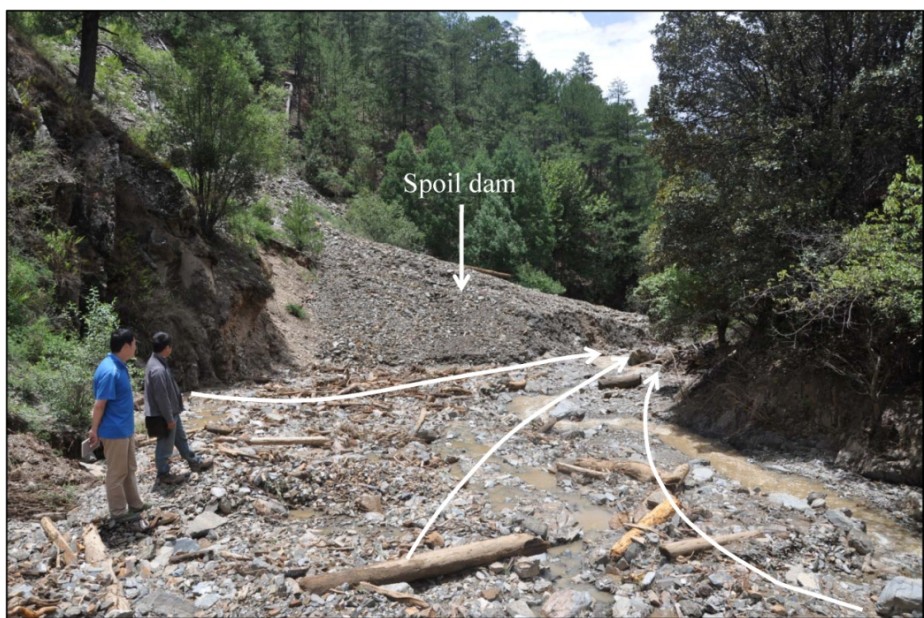

Figure 14. The main channel narrowed by spoil and spoil-induced debris flows



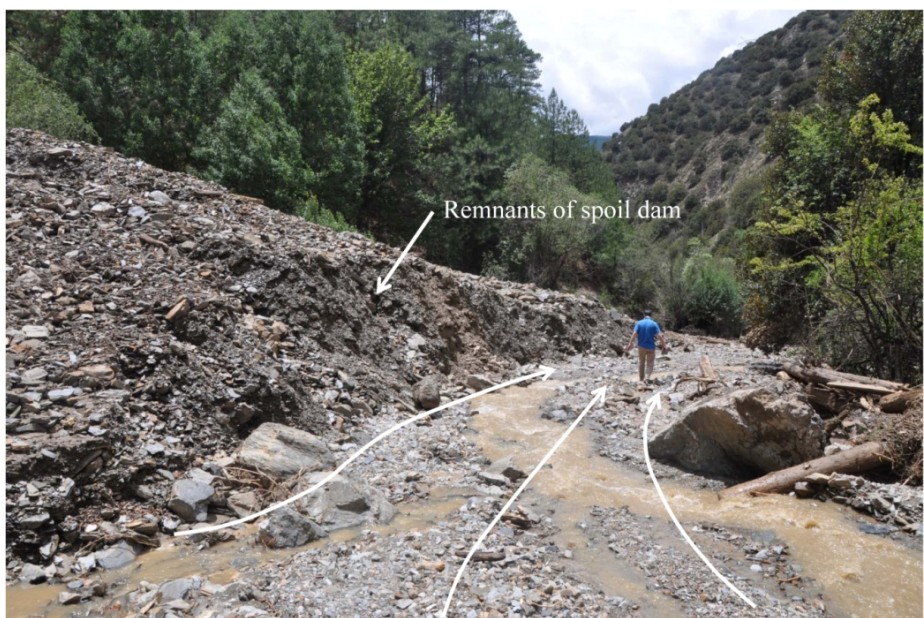


Figure 15. Remnants of a spoil dam after debris flows