# Peer review of "Multiply factors driving continual post-wildfire debris flows with varied rainfall thresholds in the Reneyong Valley, southwestern China"

_Natural Hazards and Earth System Sciences, 2017_

## Referee Comment (RC1) · Anonymous Referee #1 · 9 Jan 2018

General comments: The manuscript (ms) describes 4 debris flows in southwestern China. While the geographic setting of the study makes the paper novel, there are too many issue with presentation and quality of science, which makes the work unsuitable as a scientific publication. There is very limited data and in the cases where data is available there is not enough details on how it was used to calculated debris flow parameters such as volume. It might be that there is enough data for a scientific paper but it would need a complete reworking with much tighter focus around some key research questions/aims that are coupled with specific methods and results.

-Language is inadequate. There are many typos, grammar and spelling errors. The

lack of logical flow and the often-messy structure results in a manuscript that does not read like a scientific paper. In parts, the prose comes across more a like a layman account of what happened and not a research paper.

-Title contains grammatical errors and does not provide a clear picture of what the manuscript is a about.

-Abstract jumps into the case study without any having outlined the motivation for doing the study. The first few lines of the abstract needs to convey to the reader the purpose of the study. And the abstract end without any statements on implications. There is a report of results. Without expanding on the implications of these results the reader is left wondering "so what?".

-There are sections of the methods that indicate that data was collected, yet that data does not appear in the manuscript. For instance, on line 163 and in Figure 7 it appear that data on particle size distribution was collected. But where is the data?

-Some of the key results are reported in way that does not instill confidence in the work. For instance, in the section on Rainfall Threshold (Lines 294-320) I cannot make any sense of the results. How can the volume of debris flows be used to obtain a rainfall threshold? And in the section of debris flow deposits (line 261-281) there is insufficient details on how the volume was estimated. By stating the volumes were calculated based on " field measurement and indoor calculations" the reader does not get a sense of what was actually done.

-The conclusion does not relate back to what the study found.

Specific comments (these are some things that I picked up on but there are issue in language and structure throughout): Line 63-53: The aim to determine 'reasons for variation in rainfall threshold' appear without having 'determining thresholds' as a separate aim. Determining thresholds needs to be listed as a aim first. Line 81: The term 'plentiful sunshine' does not work for a scientific manuscript. Reword. Line 83: Reporting mean annual rainfall with a precision of 0.01mm is unreasonable. Reporting to the nearest mm would be more appropriate. Line 97-98: The statement that 'interviews indicate not debris flows in the 100 years prior to 2014' needs to be substantiated more. Do you mean no debris flows across the entire regions or in the specific catchments were the 2014 events occurred? Who was interviews and how can the authors be confident that those subjects were a good source of information on what happened in the last 100 years. Line 99-112: This section is too descriptive and the language comes across very untechnical. Line 116-117: Based on what evidence can the author say the debris flows are of high frequency? Line 154: Why is it relevant to the ms that the place is a good place for yalks and sheep to graze? If the information is not relevant, then don't include it. Line 194: I would suggest using a different word to 'sprinkled' Line 237-238: How was the volume determined? More detail needed. Line 313: Change 'Cannot' to 'Cannon' Line 313-320: I do not understand how this method works. Magnitude is a function of many other factors, not just the rainfall threshold. Please explain in more detail how thresholds can be obtained from magnitude.

---

## Referee Comment (RC2) · Anonymous Referee #2 · 11 Feb 2018

The paper deals with rainfall triggered debris flow related to wildfire.The Study area is the reneyong valley (China) where on july 2014, 4 debris flows have been triggered by heavy rainfall 3 days after a wildfire. The description of the event is very detailed and contains many hints for interpretation of the debris flow dynamics and possible further analysis. Unfortunately, the paper itself does not benefit from the quantity and quality of data and at this stage, it is not ready for publication on ISI peer reviewed journal. The innovation of the methods isn't clearly stated. For example: the volume calculation does not contain any reference.Why did you use this approach? who used it before? How different are the results you obtained compared to other methods? None of those questions has been addressed. Moreover, the structure is confusing and the language

is not clear to me.

possible insights: "Before the debris flows reached the downstream area, the patrolman (Jiuli) again found no water flows in the channel and warned the local people to escape." Why there was no water in the channel? following this observation, possible hydrological schemes of the area could be identified which could be tested in further research with geophysics.

Although a lot of work has been done in field, the paper does not contain any geomorphological map.

The authors identify a relationship between debris flow and wildfires but no historical data are used. I suggest to perform an historical research on wildfires (last 30- 50 years), debris flow activation and rainfall.

Please also note the supplement to this comment:
https://www.nat-hazards-earth-syst-sci-discuss.net/nhess-2017-390/nhess-2017-390-RC2-supplement.pdf